# The citizen's perception of a shared responsibility during the COVID-19 management: Insights from a focus group study across four European countries

L. S. Kengne Kamga[1,2]*, A. C. G. Voordouw[3], M. C. De Vries[3], S. Kemper[1,2], M. P. G. Koopmans[4], A. Timen[3,5]

1 Centre for Infectious Disease Control, National Institute for Public Health and the Environment, Bilthoven, The Netherlands, 2 Center for Infectious Disease Research, Diagnostics and Laboratory Surveillance, National Institute for Public Health and the Environment, Bilthoven, The Netherlands, 3 Athena Institute, VU University Amsterdam, Amsterdam, The Netherlands, 4 Erasmus Medical Center, Department of Viroscience, Erasmus MC, Rotterdam, and Pandemic and Disaster Preparedness Centre, Rotterdam/Delft, The Netherlands, 5 Primary and Community Care, Radboud University Medical Centre, Nijmegen, The Netherlands

* Sandra.kengne.kamga.mobou@rivm.nl

## Abstract

### Background

The World Health Organisation emphasises the importance of a whole-of-society approach to the management of health emergencies, which includes a need to involve citizens. Yet, little guidance is found on how this should occur and, more importantly, the citizen's current perspective of their role. Understanding citizens' perceptions is the first step in preparing for citizen engagement during a future pandemic.

### Methods

A qualitative research study of 16 focus groups comprising 89 participants was conducted in Finland, Slovenia, Spain, and the Netherlands. Four age-stratified focus groups in each country were organised online in the country's primary language. Inductive analysis of each focus group transcript was used to identify important themes that captured the participants' perceptions of their role in COVID-19 preparedness, response, and recovery.

### Results

Three common themes were found in the cross-country analysis. The first theme was the citizen's personal involvement, with specific references to the citizen's responsibilities and their concrete actions during COVID-19 preparedness, response, and recovery. The second theme was that of the citizen as an information receiver, with

**Data availability statement:** Due to the sensitive nature of the study, the datasets analysed during the study are not publicly available. They are archived in the National Institute for Public Health and the Environment's Centre for Infectious Disease Control 's research department, under the title of this manuscript. They are available upon reasonable request via lci-onderzoek@rivm.nl.

**Funding:** The Centre for Infectious Disease Control at the National Institute for Public Health and the Environment received funding for this research from the European Union (grant number 848096—SHARP JA—HP-JA-2018/HP-JA-2018), as part the SHARP Joint Action (https://sharpja.eu/). The funders did not play any role in the study design, data collection and analysis, decision to publish, or preparation of the manuscript.

**Competing interests:** The authors have declared that no competing interests exist.

specific references to the quantity and quality of the information received. The final theme was the relationship between the citizen and decision-makers, with specific references to the citizen's level of trust in the decision-maker, the decision-maker's communication style as perceived by the citizens, as well as the level of interaction between the citizens and decision-makers as perceived by the citizens.

## Conclusions

Generally, citizens included in this study from Finland, the Netherlands, Slovenia, and Spain felt a shared responsibility in curbing the spread of the Severe Acute Respiratory Syndrome Coronavirus-2 (SARS-CoV-2) during the pandemic. However, they emphasised the imperative need to be better informed about the likelihood of, and the developments during a pandemic in their country. Furthermore, the quality of the information provision emerged as an important theme.

---

## Introduction

The SARS-Cov-2 virus, first reported in China in December 2019, spread quickly resulting in a pandemic [1]. This rapid spread posed great challenges for governments around the globe [1]. Despite the experiences gained during past outbreaks, such as the SARS-CoV-1, Ebola, the 2009 A/H1N1 influenza pandemic, and the ZIKA outbreaks, governments and their citizens seemed insufficiently prepared for a pandemic of such magnitude. Healthcare systems were overwhelmed, resources appeared insufficient, and no vaccine was available at the pandemic outset. There was a large demand for knowledge about the virus and the possible response strategies.

To curb the spread of the SARS-CoV-2 virus, most governments introduced non-pharmaceutical interventions, with varying degrees of strictness. These ranged from hygiene measures to societal lockdowns. Such non-pharmaceutical interventions depended on citizens' engagement and required residents exhibit specific behaviours. These behaviours ranged from coughing in one's elbow to social distancing.

Pandemic management is a complex iterative process, with distinct actions required through the preparedness, response, and recovery phases. Action from governmental authorities and experts such as epidemiological surveillance and the provision of medical care, is insufficient for effective outbreak management [2]. The (potentially) affected citizens need to participate as well. The implementation of many outbreak control interventions rely on people's support and their willingness to adhere to the advice and policies put forward by health authorities [3]. The necessity of community support for outbreak management policies became increasingly clear during the 2009 H1N1 pandemic and 2014 Ebola outbreak in West Africa. The effectiveness of outbreak response interventions during both these outbreaks was hampered by citizen mistrust of health authorities, and the spread of false information [4–6].

The 2009 World Health Organization (WHO) guidance document on Pandemic Influenza Preparedness and Response [7], and the 2017 WHO Strategic Framework

for Emergency Preparedness [8] identify key principles and elements of effective, in-country health emergency prepared-ness management. The 2017 WHO Strategic Framework for Emergency Preparedness states that a coordinated, multi-sectoral and whole-of-society approach is critical for efficient preparedness [8]. It highlights the role of citizens, stating that "Community members are the first responders – and the first victims – of any emergency and, as such, essential members of the preparedness process" [8]. Consequently, citizens should be "represented in all activities around developing and implementing plans for emergency preparedness" [8]. These principles are reiterated in the more recent Regulation [EU] 2022/2371 of the European Parliament and the Council [9].

In a broader context, the importance of public engagement in health policy implementation has received increased attention in the last two decades [10]. Various arguments have been made to illustrate the benefit of public engagement in policy processes. Literature provides three rationales that motivate why citizens should be involved in policymaking. These are the substantive, the normative, and the instrumental rationales [11–15]. The substantive rationale suggests that citizens can supply new ideas and insights unknown to the experts and decision-makers [11–16]. For example, it is recognised that communities may have valuable local knowledge and social networks that are inaccessible in top-down approaches to policymaking. This chimes with the notion of collaborative governance, where it is assumed that no single party has all the knowledge and capacity necessary to deal with complex problems [17].

The normative rationale suggests that involving citizens in decisions that affect their lives, or the society they live in, is the 'right thing to do' within a democracy [11–15]. It has been argued that public engagement can increase the legitimacy of policies as policymaking processes become more transparent, increasing trust and accountability [10,18–20].

Lastly, the instrumental rationale suggests that citizens should be involved in policymaking to effectively achieve pre-defined goals [11–15]. By considering perceptions and concerns that are prevalent in a community, public authorities can devise policies that are acceptable to communities and therefore have a greater chance of being adopted [21,22].

Alongside these theories on why citizens should be involved in decision-making, literature provides various ways of describing the degree to which citizens should participate. The Arnstein [23] ladder describes a continuum of participa-tory power and consists of eight rungs. Namely, (1) citizen control, (2) delegated power, (3) partnership, (4) placation, (5) consultation, (6) informing, (7) therapy, and (8) manipulation. Charles and DeMaio [24] propose three categories of participation, namely (1) lay domination, (2) partnership, and (3) consultation. Feingold [25] discusses five degrees of participation - (1) citizen control, (2) delegated power, (3) partnership, (4) consultation, and (5) informing. More recently, the International Association for Public Participation (IAP2) introduced a model of public engagement in 2005 [26], which, again, consists of five levels, as shown in Table 1 [26].

There is also a growing body of literature on citizen participation specifically during outbreak management. Here, there is often an emphasis on compliance with non-pharmaceutical interventions. Within this context, some researchers have conducted research on individuals' disposition and character traits [27–33], such as trust in authorities, as determinants

**Table 1. The International Association for Public Participation's levels of engagement [26].**

| Level of engagement | Relationship with the community |
| --- | --- |
| Empowering | 'We will implement what you decide.' |
| Collaborating | 'We will look to you for direct advice and innovation in formulating solutions and incorporate your advice and recommendations into the decisions to the maximum extent possible.' |
| Involving | 'We will work with you to ensure that your concerns and aspirations are directly reflected in the alter-natives developed and provide feedback on how public input influenced the decision.' |
| Consulting | 'We will keep you informed, listen to and acknowledge concerns and aspirations and provide feed-back on how public input influenced the decision.' |
| Informing | 'We will keep you informed.' |

of compliance. Other researchers have investigated demographic and social characteristics [32,34–36] such as age and gender, as determinants of compliance.

Despite the body of literature on citizen participation, there is limited research on the active role of the citizens in pandemic management. Existing literature on citizen engagement emphasises the initiation from authorities. Citizens are generally described as 'receivers', rather than those who take explicit action or make decisions [37]. Hence, this study focuses on the citizen's perspective and aims to explore the understanding of citizens living in Finland, the Netherlands, Slovenia and Spain of their role in the COVID-19 pandemic management. The results capture three themes that can be used as a basic understanding for decision-making concerning citizen participation in preparedness planning.

## Study design

Facilitated focus group discussions were chosen to identify group norms and shared cultural understandings and values among participants with similar characteristics.

### Recruitment

**Country selection.** This cross-country study was conducted within the context of the European Union Joint Action on Strengthening International Health Regulations & Preparedness in the EU (EU SHARP JA). The EU SHARP JA was a European collaborative action of 26 countries and 61 partners that aimed to improve the implementation of the 2005 International Health Regulation (IHR) and EU Decision 1082/2013 on serious border threats to health [38]. One of the JA's foci was multisectoral collaboration during preparedness and response planning. The JA took place between 2019 and 2023.

All SHARP JA partners were informed of this study and were invited to participate. At the time this study was conducted, most of the individuals involved in the SHARP JA were actively involved in the pandemic management. This placed severe time and resources constraints within the JA. Partner countries who were willing and able to participate to organise the FGs were included. The aim was not to provide a comprehensive cross-country comparison, but rather to explore the perspectives of citizens living in different European countries. This pragmatic approach saw the inclusion of four countries, namely, Finland, the Netherlands, Slovenia, and Spain.

In April 2021, 16 online focus groups (FGs) were held amongst members of the general Dutch, Finnish, Slovenian, and Spanish population.

**Participant recruitment.** The aim of the recruitment was to hold an in-depth, explorative discussion with participants across different age categories from the four participating countries, rather than to yield representative samples of the populations within the four countries investigated.

Within each country, four focus groups were organised based on age stratification, i.e., 18–30, 31–45, 46–65, and 65 years plus. This saw a feasible number of participants for an online FG, whilst avoiding the potential of discussions becoming too focused on generational differences.

Various convenience sampling techniques were used to recruit participants. Participating countries had the freedom to employ their preferred recruiting method to meet possible logistical and cultural specificities. In Finland, non-governmental organisations (NGOs) working with youth, families, and pensioners sent the invitation to participate to their members. In the Netherlands, invitation letters were sent to Dutch postal addresses randomly selected from the list of addresses available on the Dutch National Postal Service's website. The invitation was also extended to the moderators' professional and personal networks to reach specific age groups. In Slovenia, the National Institute of Public Health's social media published a recruitment message, and the Slovenian moderators shared details with their personal networks to target the age groups. Spain moderators randomly selected relevant participants from a panel provided by a market investigation company. The company informed the selected panellists of their contribution and obtained their agreement to be contacted by the National Centre of Epidemiology.

**Participant country context.** The SARS-CoV-2 virus was first reported in December 2019 in Wuhan, China [39]. It quickly spread worldwide and on 30 January 2020 the WHO declared the resulting COVID-19 pandemic a public health emergency of international concern (PHEIC) [40]. By April 2021, when this study was conducted, over 761 million confirmed cases and 6.8 million deaths had been reported worldwide [41]. Europe was struggling with the pandemic's third wave, as a new variant spread rapidly [42].

In Finland, the first confirmed case was reported on 29 January 2020. The Finnish Government announced a state of emergency on 16 March 2020, and introduced non-pharmaceutical interventions, including social distancing, border controls and travel restrictions [43]. By May 2020, the spread of the virus seemed to stall, and the Finnish government decided to shift from large-scale restrictive interventions to more targeted interventions. Hence, some restrictive non-pharmaceutical interventions were retracted [44]. On 1 March 2021, the country once again declared a state of emergency as it dealt with an increasing number of cases since February 2021 [45]. Non-pharmaceutical interventions were enacted, including the closure of public premises, restrictions on group gatherings, as well as the closure of food and beverage businesses. By April 2021, Finland's population of 5.5 million [46] reported the least number of confirmed cases and deaths per million people of the four participating countries, as shown in Figs 1 and 2.

The Netherlands, with a population of 17.5 million [48], confirmed its first SARS-C0V-2 case on 27 February 2020 [49]. The Dutch government gradually introduced non-pharmaceutical interventions starting from the beginning of March 2020, resulting in what the government called an "intelligent lockdown" on 16 March 2020 [50]. During this lockdown, there was a restriction on the gathering of large groups, the implementation of 1.5m social distancing rule, and advice for home quarantine [51]. Most non-pharmaceutical interventions were relaxed in June 2020 as testing capacity increased [52], to be introduced again in September 2020 as the number of cases rose [53].

The 2 million population of Slovenia [54], had its first confirmed case reported on 4 March 2020 [55]. Non-pharmaceutical interventions were introduced between 16 and 23 March 2020, with educational institutions, public transportation, hotels, restaurants, and non-essential retail shops shut down or trading suspended [55]. During the spring of 2020, Slovenia had one of the lowest number of cases per 100,000 people in Europe. Like the other countries, interventions were initially relaxed in June 2020 [56]. However, the situation had changed by the Autumn of 2020. The number of COVID-19 cases rose drastically [57] and Slovenia's number of deaths per 100,000 inhabitants were amongst the ten highest worldwide [58]. At the time the study was conducted, the government had once again implemented non-pharmaceutical interventions [59].

With a population of 47 million [60], Spain had its first confirmed case reported on 31 January 2020 [61]. The government introduced the first non-pharmaceutical interventions on 9 March 2020 and declared a national state of emergency on 14 March 2020. The declaration was followed by a restrictive lockdown from 15 March 2020 [61], when strict non-pharmaceutical interventions were adopted [62]. These included limiting the freedom of movement within some municipalities [62]. Spain was severely affected at the beginning of the pandemic, as together with Italy [63,64], it had the highest number of COVID-19 cases and deaths. However, its relative position changed over time, as shown in Figs 1 and 2. Similarly to other countries, restrictive non-pharmaceutical interventions had been reimplemented by the time this study was conducted [65].

## Focus group guide

Researcher LSKK developed the FG guide (S1 Annex Focus group guide) using the rationales of citizen participation [11–15] and theories of the levels of citizen participation to devise open questions that captured insight on why, or to which extent citizens were involved in COVID-19 preparedness, response and recovery. The aim of the focus group guide was not to prescribe specific predetermined themes, but rather to explore concepts and themes which arose amongst participants when discussing citizen involvement during the COVID-19 pandemic.

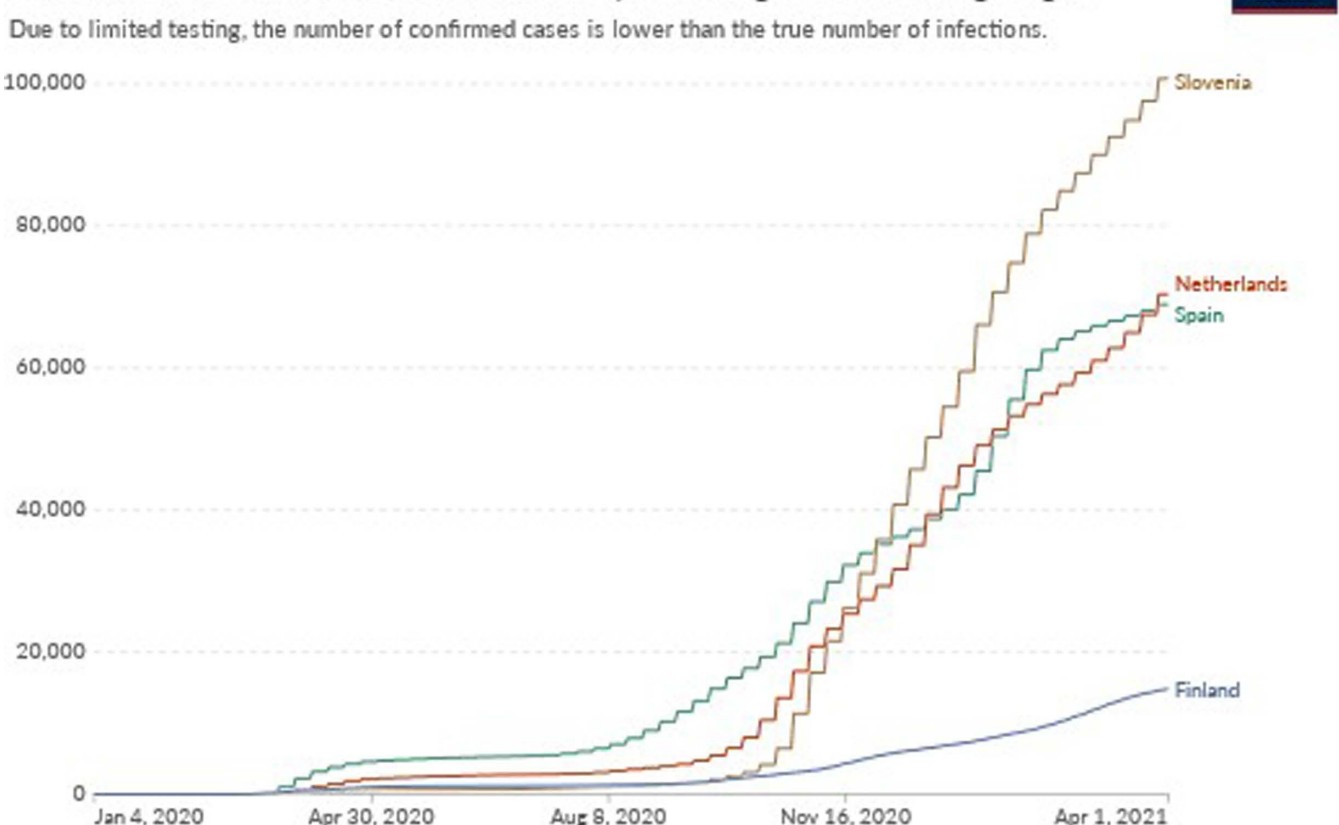

**Fig 1. Cumulative confirmed COVID-19 cases per million people.** Retrieved from https://ourworldindata.org/coronavirus#coronavirus-country-profiles [47].

The FG structure opened with introductions to the study and individuals present as well as included gaining oral consent to record the session. Five topics were then introduced, and discussion was facilitated on the role of the citizens during the distinct phases of the COVID-19 pandemic. The topics were (i) the participants' general perceptions of the pandemic, (ii) the participants' executed and expected actions before, during and after the pandemic, (iii) the participants' desire to convey their opinions or concerns with regards to the pandemic, (iv) the actors the participants believe are involved in pandemic preparedness, response and recovery in their country, and (v) which role the participants believe they should have in preparedness, response and recovery.

The FG guide was reviewed and approved by the study authors and FG moderators. The moderators were ALL and VK in the Finnish Institute for Health and Welfare, UK and AO in the National Institute of Public Health in Slovenia, CV and IL in the National Centre of Epidemiology in Spain, LSKK and VP in the Dutch National Institute for Public Health and the Environment.

FG moderators translated the FG guide into the respective languages of their countries. The translations were performed by the moderators themselves, or by translating services hired by the moderators. All translations were translated back to English to identify possible alternative interpretations of the questions. There was strong collaboration between the moderators across the different countries to clarify uncertainties and minimise biases.

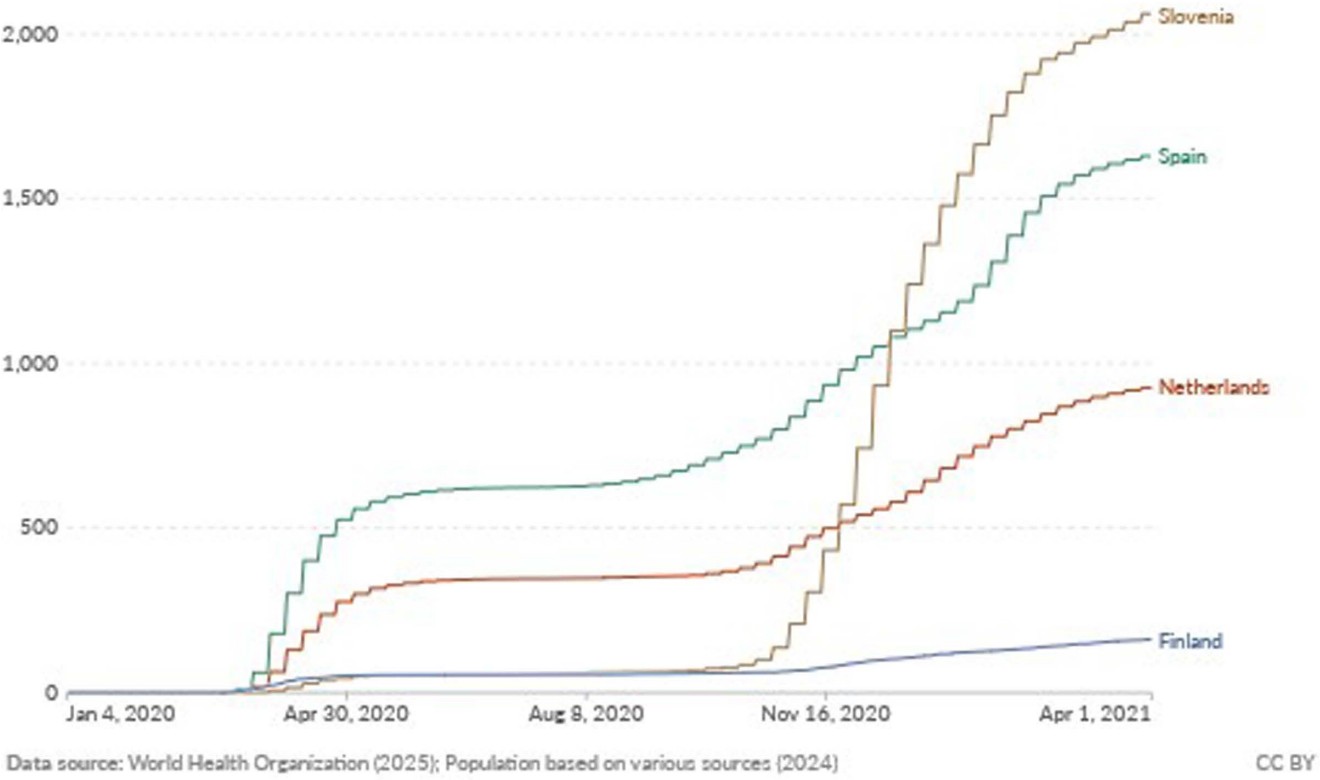

**Fig 2. Cumulative confirmed COVID-19 deaths per million people.** Retrieved from https://ourworldindata.org/coronavirus#coronavirus-country-profiles [47].

Each FG session lasted approximately 2 hours using the platform for online meetings commonly used in each institute. The focus groups were conducted in the country's main language and were recorded for transcription purposes.

## Ethics approval and consent to participate

The Medical Research Ethics Committee Utrecht (the Netherlands) assessed this study (proposal number 21/111) and judged that approval from the Central Committee on Research Involving Human Subjects was unnecessary. This assessment was shared with the Slovenian, Spanish and Finnish moderators, and was confirmed by their local ethics committee where required. The study was specifically re-approved (proposal number CEI PI 24_2021-v3) by the Instituto de Salud Carlos III Research Ethics Committee.

Written consent to participate was less practical for the online nature of the focus groups. All participants provided audio-recorded oral consent for their participation in the focus groups which is documented in the transcripts. All participants were informed of the intention to share the results when giving consent to participate.



## Analysis

Each FG was recorded and transcribed verbatim. The recordings were saved by the institutes in their respective countries. The transcriptions from Finland, Spain and Slovenia were translated into English. These were also translated back to Finnish, Spanish and Slovenian to identify possible transparent interpretations arising from the translating process. The Dutch transcriptions were kept in Dutch, as the coders (LSKK and SK) were fluent in both Dutch and English. The Dutch exemplary quotations were translated by LSKK. All transcripts were processed, anonymised, and sent to researcher LSKK for analysis.

Adopting a grounded theory methodology, the transcripts were analysed in an inductive process using the MAXQDA© 2020 software. The open coding technique was used to analyse the texts and create labels (or "codes") to categorise relevant themes. Through systematic comparison of coded texts (also called axial coding), we found emerging themes and subthemes. To ensure the reliability of the data interpretation, LSKK analysed all the FG transcripts and SK, as second coder, independently analysed 25% of the transcripts. The double-coded transcripts were discussed extensively by the coders LSKK and SK. Independently identified themes were discussed and agreed upon by the coders. The themes generated were also agreed upon by the broader research team.

The analysis and identified themes and subthemes were presented to the study moderators from the participating countries, i.e., Finland, Slovenia, and Spain, for cross-checking. It was evaluated whether the themes identified were a correct reflection of the focus groups held and whether misinterpretations due to cultural differences were present. The moderators' feedback was used to optimise the analysis. The final analysis and themes were once again presented to the study moderators for agreement. Following this, the themes were presented to individual European experts within the SHARP JA for feedback. All feedback and revision suggestions were incorporated, and a revised manuscript was signed off by the moderators. The results were reported in accordance with the Consolidated criteria for reporting qualitative research (COREQ) [66].

## Results

A total of 89 people took part in the 16 online FGs conducted across the four countries in April 2021. Four FGs were conducted in each country, based on age stratification as shown in Table 2. Each focus group had between 3 and 10 participants suitable for the online nature of the discussions.

Discussions on the five topics within the focus group guide led to the identification of three major themes and seven corresponding subthemes that described the participants' perception of their role in COVID-19 management. The major themes were (i) the citizen's personal involvement in COVID-19 pandemic management, (ii) the citizen as an information

**Table 2. The composition of focus group discussions.**

|  | Finland (n) (%) | Slovenia (n) (%) | Spain (n) (%) | The Netherlands (n) (%) | Total (n) (%) |
|---|---|---|---|---|---|
| Age categories |  |  |  |  |  |
| 18-30 years old | 8 | 5 | 8 | 4 | 25 (28%) |
| 31-45 years old | 7 | 5 | 4 | 3 | 19 (21%) |
| 46-65 years old | 10 | 4 | 7 | 4 | 25 (28%) |
| 66 years and older | 7 | 3 | 6 | 4 | 20 (22%) |
| Gender |  |  |  |  |  |
| Male | 8 (25%) | 4 (24%) | 13 (52%) | 5 (33%) | 30 (37%) |
| Female | 24 (75%) | 13 (76%) | 12 (48%) | 10 (67%) | 59 (66%) |
| Total | 32 | 17 | 25 | 15 | 89 |

receiver, and (iii) the relationship between citizens and decision-makers. These themes are elaborated upon in the following sections.

**The citizen's personal involvement in COVID-19 pandemic management**

As study participants discussed their (potential) role during the COVID-19 pandemic, an important theme that arose was that of their personal involvement in COVID-19 pandemic management. Here they focused on (i) their perceived personal responsibility in the management of the COVID-19 pandemic, and (ii) their actions during preparedness and response.

**The citizen's perceived responsibility during COVID-19 preparedness, response, and recovery.** A sub-theme that was discussed in all the focus groups across all four countries is that of the citizen's personal responsibility in the COVID-19 management. Most participants agreed that the citizen had some personal responsibility, which generally related to not infecting others and following the decision-makers' guidelines concerning non-pharmaceutical interventions. This is illustrated in the following quotation.

*"I appeal again to individual responsibility, beyond the decisions of politicians or those who have control of the rules is our own decision as people to know what we have lived and face it. There are going to be other pandemics, other situations that are going to put us all in danger, each person has the responsibility to assume the actions and decisions for their well-being and the well-being of others."* Spain 46–65 years old

In several focus groups in Finland, Slovenia, and the Netherlands, comments were made on the desired level of responsibility, and whether too much was expected from the citizen. The dynamic between the responsibility of the government and the citizens' responsibility was raised by some participants who felt that the citizens should not be expected to take on too much responsibility in respect of curbing the spread of the SARS-CoV-2 virus.

*"It might be necessary to not require so much from everyone. I feel like our society has become this place where we should all be superhumans, and young people are depressed and stressed like crazy because they're being asked to do so much."* Finland, 31–45 years old

In contrast, others felt that the governments were not giving citizens sufficient opportunities to actively participate and take responsibility in the COVID-19 response activities.

*"So, I think people are not seen in a way of being able to actively participate in curbing the virus...that is...if we're talking about responsibility. We can't assume responsibility if no one ever entrusted us with it. We were familiarized with the measures and our responsibility was to observe them."* Slovenia, 31–45 years old

**The citizen's actions during preparedness and response/recovery.** Discussions on citizen's actions during preparedness and response allowed for further insights. The sentiment of being unprepared for such a pandemic, as well as being unaware that the environment they lived in could be vulnerable to such an event, was discussed in the four Finnish FGs. This sentiment was generally shared in the other countries. Several participants in the three Spanish lower age groups participating stated they were unprepared, as they had not imagined something like this could happen to them; they had expected that pandemics are things that happen to others. Some participants from Slovenia and the Netherlands were also surprised such a pandemic could happen in their countries. They had expected the technological advances and healthcare systems in their countries would be able to manage such a public health situation.

*"What I just said, you hear things in the news about other parts of the world where it's bad. But we are the Western World, and we have the feeling that we have it figured out and that this would not happen to us. Until it was actually here, and I started to realise that people around me were asking, "What is this?""* The Netherlands, 18–30 years old

Some of the participants reported being caught unprepared for the pandemic and a few reported stockpiling products that proved to be necessary. Individual participants from Finland, Slovenia, and Spain mentioned ways to be prepared, referring specifically to having stockpiles of personal protective equipment (masks), hygiene products and toilet paper.

*"I was not prepared at all, people even had to lend me masks. At first, I had only one mask and I had to wash it, a disaster; This year we have been lucky that little by little we have been adapting. In brief, I was not prepared at all."* Spanish, 18–30 years old

With regards to the response and/or recovery phases, other than adhering to non-pharmaceutical interventions, the participants in the four countries who could suggest actions they undertook/or could undertake to support the management of COVID-19, stated that they could contribute by helping the businesses which were shut down during the lockdowns. The importance of buying locally was specifically emphasised in all countries.

### The citizen as an information receiver

The discussions on the citizen's (potential) role during pandemic management, revealed a second theme, namely the citizen as an information receiver. In general, the citizens addressed the quantity and quality of information they required as they dealt with the COVID-19 pandemic. Two subthemes further emerged, (i) the citizens feeling overloaded with information, and (ii) the citizens receiving contradictory information.

**Citizens being overloaded with information.** The presence of an information overload was discussed in the four Finnish focus groups, the youngest two Slovenian groups, and the oldest three Spanish groups. Some participants reported that this information overload left them feeling anxious and resulted in some of them disconnecting from news sources. Specific participants from the youngest and oldest Dutch focus groups commented on the perceived information chaos causing them to avoid social media and talk show channels.

*"It is true that the media are putting me in a position of great distress. I don't even read the news anymore. And it has been like this for months. I'm forced to avoid the news intentionally. On MMC, i.e., on the RTV Slovenia channel, there is news about it every five minutes. You actually visit their page to get a piece of information, but then you leave the page mentally broken."* Slovenia, 31–45 years old

Conversely, specific 46–65 years old Finnish and Slovenian participants specified that they continued to actively consume a lot of news concerning the COVID-19 pandemic.

*"So, precisely because of those worries I began following everything and once it has a hold of you, it's one of those things that just kept me fixed, so I keep listening, reading, following everything that is happening, including in education."* Slovenia, 45–65 years old

**Citizens receiving contradictory information.** Besides citizens commonly feeling overloaded with information, there was also the broadly shared sentiment that they received a lot of contradictory information. This was highlighted in all focus groups in Slovenia, Spain, and the Netherlands. The feeling that the media and politicians were not unified in providing reliable and correct information due to political and journalistic interests was discussed in the Spanish FGs. The difficulty of finding valid information was underlined by some individuals in the Slovenian 18–30 years old group.



Several respondents from the Dutch 46–65 years old group participants mentioned that the scientific basis for some of the claims made by decision-makers was unclear, as their claims later changed without explanation. For example, the case concerning the decisions regarding face masks. Furthermore, individuals in the Dutch 66 years and older focus group stated that they felt that any citizen could profile themselves as a semi-expert which was problematic.

*"In my opinion a lot of attention is being paid in the television shows, the news shows, the rubrics, and we all become semi-experts. Every girl that ever wrote a song and was able to yell in a microphone becomes an expert. That really irritates me."* The Netherlands, 66 years and older

Some of the participants in the 31–45 years old Dutch group mentioned that citizens need to use their common sense to determine where to draw the line between the information they should believe and the information they should not believe. Several participants from the Finnish 18–30 years and 66 years and older groups commented on the difficulty of finding reliable information and highlighted an issue of the media juxtaposing experts with contradictory information.

*"There's been a lot of uncertainty along the way, and especially uncertainty about which information is reliable, and how everyone will take it."* Finland, 18–30 years old

Alternatively, a general satisfaction with how local and regional coordination groups provided information was noted in the other two Finnish focus groups.

*"I also thought that I have always been very satisfied with the local coordination group, how it has taken care of the situation here in North Ostrobothnia. Recommendations have been very clear, and the validity of the recommendations has been notified, which are current recommendations and all that. Here it has worked out quite well."* Finland, 31–45 years old

**The relationship between citizens and decision-makers**

The final theme arising during the focus groups when discussing the citizen's (potential) role during the COVID-19 pandemic, is the relationship between the citizens and the decision-makers. Here three sub-themes could be identified, namely (i) the level of citizens' trust in the decision-makers, (ii) the decision-makers' communication styles as perceived by citizens, and (iii) the level of interaction between the citizens and decision-makers as perceived by citizens.

**The level of citizen's trust in the decision-makers.** In this first subtheme, there was a clear split in opinions in the four countries concerning the level of participant's trust in the decision-maker. The Dutch participants from the youngest and oldest age groups, as well as the Finnish participants from the younger three age groups generally expressed a high degree of trust in the decision-makers' knowledge and felt that the decision-makers have the best intentions for the population. A low level of trust towards decision-makers, particularly towards politicians was expressed across all four Slovenian focus groups. Some focus group participants expressed a distrust in the government's genuine intentions to protect the citizens. This was a sentiment shared by some participants across all four of the Spanish focus groups. However, there were also participants from three of the four Spanish focus groups who acknowledged the difficult tasks the decision-makers had in managing an unknown crisis.

*"I really believe that it would not do much to tell the President of the Government what needs to be done now, as some colleague has said. I do not think they are looking towards the people but looking towards themselves. I am quite disenchanted, I like politics, but I am very disenchanted."* Spain, 31–45 years old

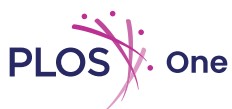

*"I really do not know if I want to tell the politicians anything, because being aware of the role that came on their heads and as much as we are not happy with the management, they have to manage an unknown disease."* Spain, 31–45 years old

**The decision-makers' communication styles as perceived by citizens.** Similar to the differences across the countries in terms of the level of citizens' trust in the decision-makers, there is also a variance in the perceptions of decision-maker's communication styles. In the Finnish group of 66 years and older, it was said that the government made suggestions on what to do. In three older Dutch focus groups there seemed to be the opinion that it was difficult for (groups of) individual decision-makers to be resolute due to the democratic nature of the country. It was suggested that the decision-makers could have communicated more firmly.

In the Slovenian focus group with 31–45 years old participants, some participants resented that decision-makers had not explained how decisions were made. It was suggested that if the decision-makers had done so, citizens might have had a better understanding of the guidelines. In the Spanish focus groups, there were once again two opinions. In the youngest Spanish focus group, the particular view that politicians selectively shared information in an infantilising manner was expressed by some individuals. On the other hand, the other individuals in the Spanish 46–65 years old group expressed the opinion that decision-makers were sharing information in a manner that encouraged the citizens to voluntarily listen to the interventions.

*"I think there has been a huge lack of sensitivity when a very large number of people have died, and we were not told what was happening. I think the problem has been that what was happening has been hidden a bit. It was preferred to treat people like small children. I think that the information has been manipulated, as other participant says, in a way or another (depending on the point of view of each person). I find it very shameful on the part of all politicians."* Spain, 18–30 years old

*"Here what you have to do is to inform the population so as they voluntarily take the measures they have to take. In fact, we are doing it more or less well."* Spain, 46–65 years old.

**The level of interaction between the citizens and decision-makers as perceived by citizens.** A final important subtheme related to the level of interaction between the citizens and decision-makers as perceived by the citizens. Some participants in the Slovenian and Finnish focus groups focused more on whether they wanted to express their opinions towards the decision-makers. Whereas the several participants in the Spanish and Dutch focus groups focused more on the decision-makers' willingness to listen to the citizens.

Across all four Slovenian focus groups, it was suggested that Slovenians are generally vocal. In the Slovenian 18–30 years old focus group, it was said that participants are not experts. Yet, it was also mentioned that one would have liked to have a dialogue with decision-makers to understand the decisions made. In the same age group, the need for a clear system that empowers citizens to state their opinions was expressed by some individuals. The expected role of the trade unions as a means for citizens to communicate with decision-makers was mentioned in the Slovenian focus group with 31–45 years old participants. However, there was the opinion that the trade unions themselves have limited dialogues with decision-makers. In the two older Slovenian focus groups, several participants criticised the behaviour of some individuals in the population who used social media or protests to criticise the decision-makers.

*"Slovenes always said that we are an intelligent and hard-working nation. And everything that this has now brought us, what all those social networks brought, in short to hide behind a pseudonym and you can do what you want and run your mouth."* Slovenia, 46–65 years old

Contrary to the Slovenian experience, most Finnish participants generally did not wish to be very vocal with their opinions during the pandemic. Individual participants from the Finnish 31–45 years, and 66 years and older focus groups explicitly mentioned that they trusted that decision-makers made decisions to the best of their abilities based on the available information, and whilst considering different perspectives. One participant from the Finnish 18–30 years old group did state that they would have liked to be heard more often, despite believing that the decision-makers were also considerate of her situation as a student. Worker's unions and personal relationships with members of parliament were mentioned as ways of communicating their opinions if they would have liked. One issue specific to the Finnish participants in the 31–45 years old, and 46–65 years old groups was that speaking up requires energy and resources, which participants who raised this issue felt they did not have.

*"I do have one Member of Parliament whose complaints I could've used to take my message forward. But there's also the fact that you do start to get fatigued, so as time passes, you don't have the energy and you just stand back and see what'll happen. The kind of personal influencing has fallen off along the way."* Finland, 46–65 years old

Generally, a negative impression of the level of interaction between citizens and decision-makers was expressed in all Spanish focus groups. The lack of belief that decision-makers truly listened to the population's problems and opinions, let alone put them above their political interests, was discussed in those groups.

*"I would like to be able to express my opinion, that they listen to me, but I get the impression that they listen to us very little. Those who have to make these decisions, especially the politicians, who are concerned about prospering, that their party works, and they do not listen to us."* Spain, 66 years and older

Contrary to those Spanish opinions, most Dutch participants from the three older age groups felt that decision-makers did listen to society in one way or another. As within the Finnish and Slovenian groups, the role of unions was mentioned. Here, there was the opinion that union group representatives did have a seat at the decision-making table. Furthermore, it was emphasised that the schools were closed in the Netherlands during the lockdown because of societal pressure. This suggested that decision-makers did listen to society. Similarly to the Finnish results, it was discussed in the youngest Dutch focus group that the student perspective was not sufficiently heard and that it would be beneficial for student representatives to have dialogues with decision-makers. Also, one participant in the Dutch 31–45 years old group stated that they felt that some people who expressed their opinions were not being listened to. On the other hand, it was also highlighted in the Dutch 46–65 years old group that there are limits to the extent that everyone can be heard. Specific participants said that policies need to be made by representatives elected by the population as the wishes of every single individual cannot be taken into consideration.

*"I find it annoying because there are a lot of people who do have an opinion and would like collaborate but are not being listening to."* The Netherlands, 31–45 years old

*"Well I thought it refer to your question that asked what do you think about the fact that the government takes the public opinion into consideration. To a certain extent I believe it is ok but also sympathise with Participant H in that we have all chosen representatives. And they define the policy. You cannot do it with, how many of us are there? Approximately 17 million people. You can't make everyone happy, at one point the policy is determined and you have to stick to it."* The Netherlands, 46–65 years old

## Discussion

During the COVID-19 pandemic, restrictive interventions requiring compliance by citizens were implemented across European countries. Hence, we aimed to explore the perception of citizens living in four European countries of their role in the COVID-19

preparedness, response, and recovery, to provide insight on how to approach the involvement of citizens in the preparedness, response, and recovery of future pandemics. The results of the multi-country focus group study, conducted in Finland, Slovenia, Spain, and the Netherlands, revealed three main themes, namely (i) the citizen's personal involvement in COVID-19 pandemic management, (ii) the citizen as an information receiver, (iii) the relationship between citizens and decision-makers.

### The citizen's perceived responsibility during COVID-19 preparedness, response, and recovery

A general sense of personal responsibility for curbing the transmission of the SARS-CoV-2 virus was expressed by most study participants across the four countries. This is in line with the outcome of several studies that looked into the degree to which citizens complied with non-pharmaceutical measures. For example, previous studies focusing on compliance with non-pharmaceutical measures during the COVID-19 pandemic showed that there was a relatively high level of compliance in the Netherlands [67,68], Finland [69], Spain [70,71], and Slovenia [72].

Furthermore, there was a heterogeneity in the views around the level of responsibility the participants in our study believed they should have. Based on the premise that citizen participation is necessary for effective outbreak management, it is noteworthy that some participants believe that they were given insufficient opportunity to participate actively and take responsibility. Some participants also felt that authorities treated them as children. Chiam et al [37] show in their systematic review of the literature published between 2004 and 2019 on community engagement for outbreak preparedness and response in high-income settings, that citizens can deal with unexpected situations and can deliberate on a range of issues. These abilities are regardless of their lack of expertise in outbreak planning and response.

The justification for the engagement of citizens provided in the study resonates most clearly with the normative rationale for citizen participation. That is, citizens seemed to think that it is what "ought to happen" in a democratic system. The citizens are the subjects of the decisions being made and are also those bearing the intended and unintended consequences of the decisions made. This is largely in line with the WHO's stance on community engagement, whose documents often refer to the citizens as those bearing the consequences of the health problem, and therefore involving them in public health emergency management is the appropriate thing to do [8].

### The citizen as an information receiver

The quantity and quality of the information the citizens received about the pandemic were commonly mentioned. On the one hand, there was the issue of being overwhelmed by the information available and/or consumed. On the other hand, there was a suggested need for clear, relevant, and correct information. Some previous studies have shown that a high proportion of European Union citizens closely followed the COVID-19 situation, suggesting a general high awareness of the COVID-19 situation (e.g., [73]). Other studies, however have noted the negative effects of large quantity of available information [74], the dramatization of the public debate [75], and rumours in the media [76] on emotions and mental health. Further studies, have reported the perceived of lack of coordination in the communication of recommendations and non-pharmaceutical measures, resulting in incoherent und difficult to interpret messages [77]. It is noteworthy however, that a study focusing on the Finnish population showed a constant level of trust in different media during April and May 2020 [69].

Our results are in line with a study conducted by Kemper et al. [11] in April 2020 (i.e., the beginning of the pandemic) in the Netherlands. The participants in that study primarily acknowledged their roles as information receivers at that time. We hypothesise that the complexity of the pandemic, the scarcity of accurate information, and the unexpected unfolding of the COVID-19 pandemic, hampered citizen participation.

Chiam et al.'s systematic review [37] also shows that citizens wish to be informed about pandemic-related issues and how to prepare for a pandemic. The citizens in the studies included in the review expressed the desire for clear, consistent, and relatable information.

This point is especially important in the pandemic context, as a review by Cipolletta et al. [78] demonstrated that knowledge of the COVID-19 was a predictor for accepting non-pharmaceutical measures. As raised by some of this study's participants, participants included in the systematic review also stressed the importance of authorities listening and providing platforms for them to provide input during pandemic planning. This could increase the community's acceptance of pandemic response interventions. This specific argument supports the instrumental rationale, which suggests that citizen participants are necessary to effectively achieve predefined goals [11–15], such as the effective implementation of non-pharmaceutical interventions.

### The relationship between citizens and decision-makers

This study shows that trust and transparency in the relationship between decision-makers and citizens are important concepts when considering the role of citizens in COVID-19 pandemic management. Our study results suggest a disparity in the relatively high level of trust in the decision-makers expressed during in the Dutch and Finnish focus groups, compared to that expressed in the Spanish and Slovenian focus groups. Previous studies have also reported a generally high level of trust amongst Finnish [79–81] and Dutch [82–84] citizens towards their governments during the COVID-19 pandemic. Other studies have reported a lower level of trust amongst Spanish [75] and Slovenian [72,85] citizens. A 2021 European Barometer survey [86] confirms the discrepancies regarding the level of trust citizens from the four countries have in their government's management of the COVID-19 pandemic. The survey showed that in 2021 68% of Dutch and 63% of Finnish citizens trusted their governments, as opposed to 20% of Spanish and 19% of Slovenian citizens who trusted their governments. The European average was 36%. Interestingly, the documentation of the association between the level of trust and level of compliance has been discrepant in literature. Some studies have documented a positive association [79,83,87] whilst others have reported no association [34].

Besides the issue of trust, there was also the issue of the general perception of how decision-makers communicated with the citizens. In our study, a relatively negative perception was articulated in the Slovenian and Spanish focus groups. A tendential negative perception of governmental response and/or crisis communication style has also been documented in previous studies conducted in Spain [75,88] and Slovenia [89]. This resonates with findings in Chiam et al.'s literature review [37]. Similarly to this study, participants included in the review's literature expressed concerns about the whether the information provided by authorities was complete, and whether decisions made were primarily for political gain. The review also highlighted that some citizens wished for authorities to treat them as adults who can deal with complex information.

Some participants included in this focus group study did express the need for a system within which citizens could provide feedback to receptive decision-makers for those who wished to express their opinions. There do seem to be possible effective platforms for citizen engagement during PHEs. For example, some initiatives such as those reported by Mouter et al. (2020) [90], who used an approach named the Participatory Value Evaluation (PVE) method during the COVID-19 pandemic. They presented citizens with different policy options as well as the contexts within which the policies needed to be developed. Based on the information provided, citizens were given the opportunity to inform policymakers of their preferred policy options. They describe PVE as a novel participatory approach, besides existing approaches such as mini-publics, referenda, opinion-polls, and participatory budget. Mounter et al. [90] suggest that the advantages of a PVE include the fact that organising and participating in one does not require a lot of the citizen's time or extensive (sub) national logistical organisation. Furthermore, it provides the possibility for citizens to gain a greater awareness and understanding of the variety of policy options and the contexts within which those policies are developed.

Given the outcome of our study, the PVE method or another participatory approach might have been welcomed by some of the study participants. However, it is noteworthy that our study results suggest that the majority of citizens do not wish to be part of the decision-making process itself. When considering previous literature of the optimal levels of citizen participation, our study outcomes suggests that the majority of citizens do not wish for an empowered role, as defined by

International Association for Public Participation's (IAP2) model [26]. They do not want a true participation, where citizens have a shared responsibility in decision-making, have dominant power, and can control the decision-making [23].

These results are in line with Litva et al.'s [91] study on the public's perception of its role in healthcare decision-making. The participants of that study wished for some engagement but did not desire a partnership. Some of Latvia et al.'s [91] study participants highlighted that they are not experts, or criticised some individuals for acting as semi-experts. This suggests the citizens felt they did not necessarily have the required expertise to responsibly make certain decisions. Furthermore, participants in Litva et al.'s [91] study wished for mechanisms to be put in place that ensure their voices are not only heard but also that their opinions are implemented in decision-making.

The Lowndes et al. [92] focus group study on the trends in public participation from a citizen's perspective provides reasons for why citizens do not participate, three of which are relevant to our results. One reason identified by Lowndes et al. [92] is the citizen's view of the local authority; notably a negative view may deter them from participating. Within our results, there is an emphasis on the relationship between the citizen and decision-makers. Although there seemed to be general trust in the government's decisions in the Dutch and Finnish focus groups, participants in Spanish and Slovenian focus groups were more critical of the ability of the decision-maker to put the population's best interests above their own. Yet, there were no clear differences in the participant's willingness to engage with the decision-makers across the four focus countries.

Another reason provided by Lowndes et al. [92] is that citizens may be unaware of the opportunities they have to participate. The COVID-19 pandemic was unprecedented, and our results showed that many citizens were unaware of the possibility of this happening in their country. Participants recognised the benefits of decision-makers engaging with citizens, such as encouraging citizens to follow guidelines and understanding the realities lived by citizens. However, participants did not comment on the potential advantages of involving citizens in the decision-making process to improve the quality of decisions made, as the substantive rationale for citizen participation suggests [11,12]. Participants seemed unaware of the added value they could bring to the decision-making table. Using a whole-of-society approach, where multisectoral collaboration, including community representatives, is required to deal with societal problems, we propose pandemic preparedness plans need to define strategies to actively involve citizens.

A third reason identified by Lowndes et al. [92] is that citizens are unlikely to participate if they expect nothing will be done with their contribution. Unlike the Dutch participants who felt that the decision-makers did change decisions made based on societal pressure, most participants from Spain did not believe decision-makers listened to the population's problems and opinions. The Slovenian participants felt that there was no system in place that empowered the citizens' opinions. Conklin et al.'s [93] systematic review also documented the scarcity of evidence of the impact of citizen participation in health care policy. Therefore, we believe this topic is worth further research as existing international documents lack clarity on the citizen's exact role, mandate and expected activities during the distinct phases of PHEs, plus guidance on how to measure their impact. For this reason, we suggest those responsible for PHE management should actively contact citizens and aim to collaborate with them to clearly define the citizen's roles and responsibilities. An acknowledgement of the difficulty of the tasks with consideration of individual and group preferences is paramount.

## Strengths and limitations

We believe that to work towards citizens being partners in pandemic management as stipulated in whole-of-society documents, it is important to understand which roles citizens see for themselves currently, and which roles citizens think they should have. Key in this study's strengths is that it was conducted from a citizen perspective amid the pandemic allowing participants to actively reflect on aspects of the pandemic whilst they lived through it. This made the probability of recall bias low.

Another strength is the fact that the study was conducted across four European countries with different geographical, cultural, lingual, economic, and historical backgrounds. The themes described in the results were common across all four



countries, suggesting the outcome could be valuable within the individual country contexts as well as within the European context. However, it is important to acknowledge that our study sample is not necessarily representative of the population in the four countries investigated and that citizens are not a homogeneous group of people. Hence, policies designed for citizen engagement during PHE management, should take national and subnational context specific factors, such as cultural attitudes, political systems, and health care systems, into consideration.

In terms of study limitations, we acknowledge the dynamic nature of the pandemic in terms of variable epidemiology and non-pharmaceutical interventions implemented across the four countries. This study was conducted during the third wave of the pandemic. Conducting this study during another period of the pandemic could have possibly resulted in the accentuation of other themes. Furthermore, conducting the study online rather than in a face-to-face setting may have resulted in a different interaction among participants, and between the moderators and participants. This may have potentially had an impact on the discussions and hence the themes identified.

We also recognised limitations in the different recruitment techniques employed by participating countries, as well as the possibility of local nuances being lost during translation processes, and during analysis by individuals from one country. To pragmatically mitigate these potentials, the results were crosschecked with the moderators from all participating countries, as described in the study design.

Finally, based on the introductions held during the focus groups and the interactions during the focus group sessions, it became apparent that individuals from socially disadvantaged groups were not well represented. The lack of the perspective of this heterogeneous group is a potential limitation. We also recognise that citizens that participate in such a study may be more likely to be interested in sharing their opinion about this topic, and may even be likely to share socially desirable responses. Yet, we feel we were able to capture a range of perspectives, as opinions differed across countries and age categories and were at times critical of governmental institutions.

### Future research

Our study's findings across four European countries can serve as input for further in-depth studies of the citizen's (potential) role during pandemic management. Firstly, we urge for further qualitative research on this topic, preferably with a stratified sampling method, to include a more diverse sample. The deliberate inclusion of citizens with different socioeconomic backgrounds, beliefs, and lifestyles may allow for a more comprehensive exploration and understanding of the themes identified in this particular study. Furthermore, emphasis can also be placed on designing concrete and evidence-based pandemic management communication policies, as well as context-specific participatory methods for citizen engagement.

Secondly, we encourage further quantitative research with larger sample sizes in the four countries, and in additional European countries, to evaluate the (sub-) themes which have been identified in the study within a more representative sample of the populations within the investigated countries.

### Conclusion

The study showed that, in one way or another, study participants felt a shared responsibility in curbing the spread of the SARS-CoV-2 virus. Some participants limited this responsibility to compliance with the restrictive non-pharmaceutical interventions put in place. Some wished to receive information of good quality, and some wanted to engage with decision-makers. There was, however, little perceived need to participate in decision-making itself. This study suggests that the citizens included in this study have a desire to be informed about the possibility of a pandemic taking place in their country, as well as receiving quality information in a suitable manner concerning their possible roles during pandemic management. This will create a better understanding of the concrete actions citizens can undertake during pandemic preparedness and response. Understanding citizen's perceptions is the first step in preparing for citizen engagement during a future pandemic.

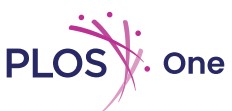

## Supporting information

**S1 Annex. Focus group guide.**
(PDF)

**S2 Annex. Translation non-WMO Statement.**
(PDF)

## Acknowledgments

We would like to thank A.L. Lohiniva, V. Härmä and O. Karvonen from Finland, C. Varela Martínez, I. León and L. Duque from Spain, U. Kolar and A. Orehek from Slovenia, and V. Peerbooms from the Netherlands for contributing to and translating the focus group guide, recruiting, and moderating the focus groups, translating the transcripts, and reviewing the manuscript. We also thank K. Dancey for editing the final manuscript. We would also like to thank all study participants from Finland, Slovenia, Spain, and the Netherlands who actively took part in the focus groups.

## Author contributions

**Conceptualization:** L. S. Kengne Kamga, A. C. G. Voordouw, M. P. G. Koopmans, A. Timen.

**Data curation:** L. S. Kengne Kamga.

**Formal analysis:** L. S. Kengne Kamga, S. Kemper.

**Methodology:** L. S. Kengne Kamga.

**Project administration:** L. S. Kengne Kamga.

**Supervision:** A. C. G. Voordouw, M. C. De Vries, M. P. G. Koopmans, A. Timen.

**Writing – original draft:** L. S. Kengne Kamga.

**Writing – review & editing:** L. S. Kengne Kamga, A. C. G. Voordouw, M. C. De Vries, S. Kemper, M. P. G. Koopmans, A. Timen.

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
