## [Decision Letter · Decision Letter 0]

25 Jul 2024

PONE-D-24-07236The perceptions amongst European citizens of shared responsibility during the COVID-19 management: A focus group study across four European countriesPLOS ONE

Dear Dr. Kengne Kamga,

Thank you for submitting your manuscript to PLOS ONE. After careful consideration, we feel that it has merit but does not fully meet PLOS ONE’s publication criteria as it currently stands. Therefore, we invite you to submit a revised version of the manuscript that addresses the points raised during the review process.

===Many thanks for your interesting contribution. There are major concerns regarding your paper which would be necessary to be addressed in a substantial revision. Please see both reviews attached and address their concerns point by point. Additionally, there are also major concerns regarding the following issues: A) Please contextualize the comparion of the empirical material/Findings with regard to each country's situation, and the material to be found, and how do you justify/ratioonalize the selection iof these countries? it seems a bit random.B) Please add in the methodology section a much more detailed description of your coding methdology, used codes, provide examples for your coding strategy (eg. as suppl. material) and desecrite better the coding process - where deductive, where inductive... . How can a Dutch scientists code properly and interprete data from Slovenia without knowing cultural codes, contexts, background? How did you solve this issue? C) The ethical concept of participation presented in the introduction part is very general and formal and not sufficiently linked to current discussion of participation in healh and science politics since. It would be worth to dive a much deeper in the last 20 years of discussion about citizen participation in science and health politics as well as - perhaps more important - the relation of participation to various understandings of democracy and of science - and how this might differ in the respectives countries you refer to... It seems necessary to provide here a more updated perspective but also to reconsider then perhaps some of your analysis, especially regarding differences between countries should be consideredD) There exist a large! body of studies, also qualitative and quantitaive studies that have also comparetively studied public perception of COVID 19 pandemic measurements in European countries or in comparison of Europe/Transatlantic/Global South. Topics of these studies refer also to very related things such as responsibilities, solidarity, injustices, and citizen's burden or more general the role or non-participation/exclusion of vulnerable groups. It seems necessary to embedded your results much more in this large body of literature to ensure that your findings are sufficiently interpreted and contextualized. We would be happy to recieve a substantial revieisions with a addtional letter to the reviewers.  Please upload a version with trackchanges and onw where inly chnages parts are highlighted, but without track changes.

Please submit your revised manuscript by Sep 08 2024 11:59PM If you will need more time than this to complete your revisions, please reply to this message or contact the journal office at plosone@plos.org . Please include the following items when submitting your revised manuscript:

We look forward to receiving your revised manuscript.

Kind regards,

Silke Schicktanz

Academic Editor

PLOS ONE

2. In the online submission form you indicate that your data is not available for proprietary reasons and have provided a contact point for accessing this data. Please note that your current contact point is a co-author on this manuscript. According to our Data Policy, the contact point must not be an author on the manuscript and must be an institutional contact, ideally not an individual. Please revise your data statement to a non-author institutional point of contact, such as a data access or ethics committee, and send this to us via return email. Please also include contact information for the third party organization, and please include the full citation of where the data can be found.

Reviewers' comments:

Reviewer's Responses to Questions

**Comments to the Author**

1. Is the manuscript technically sound, and do the data support the conclusions?

Reviewer #1: Yes

Reviewer #2: Yes

2. Has the statistical analysis been performed appropriately and rigorously? 

Reviewer #1: N/A

Reviewer #2: N/A

3. Have the authors made all data underlying the findings in their manuscript fully available?

Reviewer #1: No

Reviewer #2: Yes

4. Is the manuscript presented in an intelligible fashion and written in standard English?

Reviewer #1: Yes

Reviewer #2: Yes

5. Review Comments to the Author

Reviewer #1: This is a sound, well written and informative paper. The introduction is excellent, as it provides a quick but instructive theoretical framework on public engagement in public health policy and health emergency. The methods are appropriate and the conclusions properly follow the findings. I also think the methods and the international approach make this piece novel and attractive for the readership of PlosONE. I personally have enjoyed reading it.

I have two major concerns. If addressed, I think the paper would be an excellent contribution.

First, the current version lacks context information about the countries where the study was conducted. To say that "At that time, the COVID-19 pandemic was ongoing" (130) is insufficient, in my view. To facilitate the interpretation of the results, and to enable meaningful international comparison, I suggest adding a section where some basic data of each country regarding the Covid-19 pandemic, including: Phase of the outbreak, number of deaths attributable to Sars-CoV-2, non-pharmaceutical measures that were being adopted in each territory at the time of focus groups, public engagement initiatives in each country.

Second, and this may be less problematic: as it is often the case in qualitative studies, it is not clear to me whether the main themes and sub-themes that structure the results and drive the discussion simply "arose" (254), "emerged" (202) and were "found"(43, 202) and "identified" (210), or rather induced or facilitated by the FG guide. A way to show the extent to which the research team was "surprised" by unexpected results (or rather confirmed their results expectations), it would be useful to explicitly emphasize the closeness or distance between the themes proposed in the FG guide, and the themes resulting from the analysis.

Congrats for a great work and I hope this helps to improve an already sound and informative manuscript.

Reviewer #2: The study "The perceptions amongst European citizens of shared responsibility during the COVID-19 management: A focus group study across four European countries" presents an impressive variety of focus group interviews with citizens from the Netherlands, Spain, Slovenia, and Finland. The study's methodology is clear and seems valid, and the rationale is clearly presented. I am grateful for the opportunity to review this interesting manuscript. In the following, I present some suggestions for further improving the paper.

1. The paper (primarily the discussion) could benefit from a more thorough and nuanced embedding in the breadth of published empirical literature about citizen perceptions of the pandemic. For instance, trust in government was measured in quantitative surveys in many countries during the pandemic, and knowing to what extent citizens in the various countries trusted governments could be helpful to assess to what extent the focus groups represent majority opinions in that regard (mainly because the authors report distinct differences among countries in that regard).

Moreover, people's perceptions of responsibility, solidarity, and information gathering were investigated in similar European settings – including qualitative studies – and directly speak to this study. Thus, the authors should revisit their statement that their study was "the first qualitative study that collects information from a variety of European countries to explore the citizen's perceptions of their role during a public health emergency as the emergency unfolds".

2. Some contextual information on the four participating countries and the time of the interviews would help interpret the findings (to help explain, for instance, the different perceptions regarding trust and citizen-policymaker interactions). For instance, the political system and the general political reactions to the pandemic. The authors could report, for instance, the oxford stringency indices for the included countries over time.

3. Relatedly, the interviews were held during the pandemic, which surely influenced participants' perceptions. If the interviews were held a few months earlier or later, perceptions might have differed. Giving some background on the stage of the pandemic and what that meant for the study findings would be helpful.

4. Why were these four countries selected for this study? What are the concrete similarities and differences among them? Other European countries – e.g. Switzerland, where citizens voted on COVID-19 legislation several times during and after the pandemic – might have portrayed more variable and, thus, nuanced perceptions of citizen participation in Europe due to different underlying political systems and contexts.

5. A minor point: It is not directly clear how the themes in the findings speak to the overall topic (especially the topic of information gathering). An explicit introductory sentence might be helpful here.

6. I recommend some thorough English language editing – the language is understandable but should be improved in some instances before publication.

7. Methodological comments:

- Doing online focus groups can be challenging. What difficulties did the moderators encounter and how were they addressed? (e.g., technical problems, limited visual cues, interruptions, imbalances in how much participants contributed, etc.)

- How were the transcripts translated, and how did the authors ensure to minimize or make transparent interpretations occurring because of translations?

6. PLOS authors have the option to publish the peer review history of their article (what does this mean? ). If published, this will include your full peer review and any attached files.

**Do you want your identity to be public for this peer review?** For information about this choice, including consent withdrawal, please see our Privacy Policy .

Reviewer #1: **Yes: ** David Rodríguez-Arias

Reviewer #2: **Yes: ** Bettina M. Zimmermann

---

## [Author Response · Author response to Decision Letter 1]

5 Oct 2024

05-10-2024

Dear Professor Schicktanz,

We would like to thank you and the reviewers for providing feedback to improve the quality of our manuscript, entitled, “The perceptions amongst European citizens of shared responsibility during the COVID-19 management: A focus group study across four European countries.

The research team has carefully considered and discussed all comments and the following major changes have been made:

(1) we have introduced the topic of citizen participation in public policy in more depth and referenced more recent literature in the introduction

(2) we have provided further detail about country selection and country contexts in the design section

(3) we have further embedded our outcome in recent relevant literature.

The following pages provide point-by-point responses to all comments, indicating corresponding changes in the manuscript and where to find them. All significant changes have been marked. All authors have seen and approved the revised manuscript.

We hope this revised manuscript can be accepted for publication in PLOS ONE.

Yours sincerely,

Sandra Kengne Kamga

On behalf of the co-authors

Academic editor’s comments

===Many thanks for your interesting contribution. There are major concerns regarding your paper which would be necessary to be addressed in a substantial revision. Please see both reviews attached and address their concerns point by point. Additionally, there are also major concerns regarding the following issues:

Response

Thank you for considering our manuscript and identifying major concerns. Here below we provide information on how we have addressed those concerns.

Comment A

A) Please acontextualize the comparion of the empirical material/Findings with regard to each country's situation, and the material to be found, and how do you justify/rationalize the selection of these countries? it seems a bit random.

Response to comment A:

To address these concerns, we have expanded on the country selection within the study design section. The selection of the countries was necessarily pragmatic, driven by the resources available within the European Union SHARP Joint Action (JA) at the time the study was conducted.

The study design section now contains additional contextual information on the COVID-19 country situation in the four participating countries in April 2021.

Comment B

B) Please add in the methodology section a much more detailed description of your coding methdology, used codes, provide examples for your coding strategy (eg. as suppl. material) and desecrite better the coding process - where deductive, where inductive... . How can a Dutch scientists code properly and interprete data from Slovenia without knowing cultural codes, contexts, background? How did you solve this issue?

Response to comment B:

We have added more detail on the coding methodology, highlighting that the process was purely inductive.

We acknowledge the concern regarding the Dutch researchers’ ability to appropriately code and analyze the data from other countries. We chose to conduct this study using the resources available within the Joint Action at the time. Unfortunately, there was a limitation in moderators and experts from other Joint Action countries’ availability to participate in the analysis. Nonetheless, the analysis and results were presented to the moderators from the participating countries. Their feedback was used to optimize the analysis. The final analysis and results were once again shared with the moderators, plus all individuals participating in the Joint Action for feedback. In this manner, we attempted to minimize any misinterpretations arising from differing cultural and contextual codes and lived experiences. We have again communicated this issue revealed in the peer review process with the moderators from the other three participating countries. No objections were made and no changes in the analysis process undertaken was deemed necessary. Nonetheless, we have addressed this issue to the study’s limitations.

Comment C

C) The ethical concept of participation presented in the introduction part is very general and formal and not sufficiently linked to current discussion of participation in healh and science politics since. It would be worth to dive a much deeper in the last 20 years of discussion about citizen participation in science and health politics as well as - perhaps more important - the relation of participation to various understandings of democracy and of science - and how this might differ in the respectives countries you refer to... It seems necessary to provide here a more updated perspective but also to reconsider then perhaps some of your analysis, especially regarding differences between countries should be considered

Response to comment C

Thank you for pointing this out and your suggestion. The introduction has been expanded upon to further elaborate on more recent literature on citizen participation in health policies.

Comment D

D) There exist a large! body of studies, also qualitative and quantitaive studies that have also comparetively studied public perception of COVID 19 pandemic measurements in European countries or in comparison of Europe/Transatlantic/Global South. Topics of these studies refer also to very related things such as responsibilities, solidarity, injustices, and citizen's burden or more general the role or non-participation/exclusion of vulnerable groups. It seems necessary to embedded your results much more in this large body of literature to ensure that your findings are sufficiently interpreted and contextualized. We would be happy to recieve a substantial revieisions with a addtional letter to the reviewers.

Response to comment D

We have attempted to imbed our results in more recent literature. Specifically, adding the outcomes of some studies conducted during the COVID-19 pandemic, plus a review on citizen participation during outbreaks that considered literature conducted prior to the pandemic.

Please upload a version with trackchanges and onw where inly chnages parts are highlighted, but without track changes.

2. In the online submission form you indicate that your data is not available for proprietary reasons and have provided a contact point for accessing this data. Please note that your current contact point is a co-author on this manuscript. According to our Data Policy, the contact point must not be an author on the manuscript and must be an institutional contact, ideally not an individual. Please revise your data statement to a non-author institutional point of contact, such as a data access or ethics committee, and send this to us via return email. Please also include contact information for the third party organization, and please include the full citation of where the data can be found.

Response

The statement has been revised. It now reads, “Due to the sensitive nature of the study, the datasets analysed during the study are not publicly available. They are archived in the National Institute for Public Health and the Environment’s Centre for Infectious Disease Control ‘s research department, under the title of this manuscript. They are available upon reasonable request via lci-onderzoek@rivm.nl.”

The annexes included have been sited according to the guidelines and have been included at the end of the manuscript.

Review Comments to the Author

Reviewer #1 comments

Comment 1

First, the current version lacks context information about the countries where the study was conducted. To say that "At that time, the COVID-19 pandemic was ongoing" (130) is insufficient, in my view. To facilitate the interpretation of the results, and to enable meaningful international comparison, I suggest adding a section where some basic data of each country regarding the Covid-19 pandemic, including: Phase of the outbreak, number of deaths attributable to Sars-CoV-2, non-pharmaceutical measures that were being adopted in each territory at the time of focus groups, public engagement initiatives in each country.

Response to comment 1

Thank you for your compliments and this suggestion. To address this specific concern, we have provided contextual information on the COVID-19 situation in the four participating countries in April 2021, when the focus groups were conducted.

Comment 2

Second, and this may be less problematic: as it is often the case in qualitative studies, it is not clear to me whether the main themes and sub-themes that structure the results and drive the discussion simply "arose" (254), "emerged" (202) and were "found"(43, 202) and "identified" (210), or rather induced or facilitated by the FG guide. A way to show the extent to which the research team was "surprised" by unexpected results (or rather confirmed their results expectations), it would be useful to explicitly emphasize the closeness or distance between the themes proposed in the FG guide, and the themes resulting from the analysis.

Response to comment 2

We have added text to the Study design section that explicitly emphasizes that the focus group guide included predominantly broad open questions, which would allow for the focus group participants to steer the course of the discussions. The broad open questions were, (i) the participants’ general perceptions of the pandemic, (ii) the participants’ executed and expected actions before, during and after the pandemic, (iii) the participants’ desire to convey their opinions or concerns with regards to the pandemic, (iv) the actors the participants believe are involved in pandemic preparedness, response and recovery in their country, and (v) which role the participants believe they should have in preparedness, response and recovery.

Given the explorative and multi-country nature, no themes or opinions were proposed in the focus group guide. An analysis of the focus group transcriptions confirmed that moderators withheld from prompting discussions of specific themes or concepts that participants themselves did not raise.

Reviewer #2 comments

Comment 1

The paper (primarily the discussion) could benefit from a more thorough and nuanced embedding in the breadth of published empirical literature about citizen perceptions of the pandemic. For instance, trust in government was measured in quantitative surveys in many countries during the pandemic, and knowing to what extent citizens in the various countries trusted governments could be helpful to assess to what extent the focus groups represent majority opinions in that regard (mainly because the authors report distinct differences among countries in that regard).

Moreover, people's perceptions of responsibility, solidarity, and information gathering were investigated in similar European settings – including qualitative studies – and directly speak to this study. Thus, the authors should revisit their statement that their study was "the first qualitative study that collects information from a variety of European countries to explore the citizen's perceptions of their role during a public health emergency as the emergency unfolds".

Response to comment 1

We have attempted to imbed our results in more recent literature. Specifically, we have added the outcomes of some studies conducted during the COVID-19 pandemic, plus a review on citizen participation during outbreaks that considered literature conducted prior to the pandemic.

The sentence "the first qualitative study that collects information from a variety of European countries to explore the citizen's perceptions of their role during a public health emergency as the emergency unfolds" has been removed.

Comment 2

Some contextual information on the four participating countries and the time of the interviews would help interpret the findings (to help explain, for instance, the different perceptions regarding trust and citizen-policymaker interactions). For instance, the political system and the general political reactions to the pandemic. The authors could report, for instance, the oxford stringency indices for the included countries over time.

Response to comment 2

To address this specific concern, we have provided contextual information on the COVID-19 situation in the four participating countries in April 2021, when the focus groups were conducted.

Comment 3

Relatedly, the interviews were held during the pandemic, which surely influenced participants' perceptions. If the interviews were held a few months earlier or later, perceptions might have differed. Giving some background on the stage of the pandemic and what that meant for the study findings would be helpful.

Response to comment 3

The study design section now expands on the COVID-19 pandemic generally and in the context of the four participating countries.

Under the Strengths and limitations section we discuss the possibility that the themes identified may have been different if the study were to be conducted in a different phase of the COVID-19 pandemic.

Comment 4

Why were these four countries selected for this study? What are the concrete similarities and differences among them? Other European countries – e.g. Switzerland, where citizens voted on COVID-19 legislation several times during and after the pandemic – might have portrayed more variable and, thus, nuanced perceptions of citizen participation in Europe due to different underlying political systems and contexts.

Response to comment 4

To address these concerns, we have provided additional information on the country selection in the study design section. The selection of the countries was a pragmatic one based on resources available within the European Union SHARP Joint Action (JA) at the time the study was conducted.

Furthermore, we have referenced some studies from other participating countries, which seem to support our outcomes.

Comment 5

A minor point: It is not directly clear how the themes in the findings speak to the overall topic (especially the topic of information gathering). An explicit introductory sentence might be helpful here.

Response to comment 5

We have added an introductory sentence to the individual themes showing their relationship to the overall topic.

Comment 6

I recommend some thorough English language editing – the language is understandable but should be improved in some instances before publication.

Response to comment 6

Thank you for the suggestions. We have made textual changes that we believe it will improve the readability of the text.

Comment 7

Methodological comments:

- Doing online focus groups can be challenging. What difficulties did the moderators encounter and how were they addressed? (e.g., technical problems, limited visual cues, interruptions, imbalances in how much participants contributed, etc.)

- How were the transcripts translated, and how did the authors ensure to minimize or make transparent interpretations occurring because of translations?

Response to comment 7

We did not face any problems when conducting the online focus groups, having had experience in conducting online groups for an earlier study during the COVID-19 pandemic.

However, in the strengths and limitations section we recognise the possible impact of the online setting on interaction amongst moderators, and between moderators and participants.

Fuller information is now included in the study design section describing how the focus group guides were translated to the participating countries’ respective languages and then back translate

---

## [Decision Letter · Decision Letter 1]

13 Nov 2024

PONE-D-24-07236R1The perceptions amongst European citizens of shared responsibility during the COVID-19 management: A focus group study across four European countriesPLOS ONE

Dear Dr. Kengne Kamga,

Thank you for submitting your manuscript to PLOS ONE. After careful consideration, we feel that it has merit but does not fully meet PLOS ONE’s publication criteria as it currently stands. Therefore, we invite you to submit a revised version of the manuscript that addresses the points raised during the review process.

We look forward to receiving your revised manuscript.

Kind regards,

Silke Schicktanz

Academic Editor

PLOS ONE

**Additional Editor Comments:**

Dear authors,

we appreciate your revision efforts, but reviewer 1 has some concerns which I also share. Itthat you describe your study design according to main standards of qualitative focus group research, including giving detailed information about socio-demographics (incl. education, gender, profession, perhaps Covid-infection: affected vs. non.affectedness). Also problematic generalizations as outcome of a qualitative study should be avoided. Please replace Euroipean by naming only te countries you have examined (in the title, abstract, introduction, result and discussion parts). Discuss then more closely your findings for the respectives countries in comparison to other existing qualitative and quantitative studies of these countries (Trust-related issues into health sciences, governance, ). You can also discuss your results of your 4 countries with studies that have been undertaken in other countries but please make sure you compare apples with apples...

As qualitative studies help to buil hypothesis but not generalized sztatement, I also recommend that you apply a more reflective discussion style and develop rather an outlook part, how to test your hypothesis in future studies.

We are looking forward to recieve a revised vrsion of your manuscript

Sincerely, Yours Silke Schicktanz

Reviewers' comments:

Reviewer's Responses to Questions

**Comments to the Author**

1. If the authors have adequately addressed your comments raised in a previous round of review and you feel that this manuscript is now acceptable for publication, you may indicate that here to bypass the “Comments to the Author” section, enter your conflict of interest statement in the “Confidential to Editor” section, and submit your "Accept" recommendation.

Reviewer #1: All comments have been addressed

Reviewer #2: (No Response)

2. Is the manuscript technically sound, and do the data support the conclusions?

Reviewer #1: Yes

Reviewer #2: Partly

3. Has the statistical analysis been performed appropriately and rigorously? 

Reviewer #1: Yes

Reviewer #2: N/A

4. Have the authors made all data underlying the findings in their manuscript fully available?

Reviewer #1: (No Response)

Reviewer #2: Yes

5. Is the manuscript presented in an intelligible fashion and written in standard English?

Reviewer #1: (No Response)

Reviewer #2: Yes

6. Review Comments to the Author

Reviewer #1: It seems to me that the authors have taken seriously my remarks and, according to their answer, the manuscript has improved sufficiently to be published.

I really appreciate the effort in providing context for each participant country phase of the pandemic. I also appreciate the details given on the methodology and coding process.

In sum, I think this is a great contribution, that has high value in terms of understanding a substantive question: what kind of expertise and legitimacy has the public in engaging with public-health policy decision-making? This paper analyses this problem in depth and is extremely informative.

I think it should be published.

Reviewer #2: Thank you for addressing my comments and suggestions. However, I still have some concerns that I would like to share:

1. The vast amoung of literature on citizen perceptions on regulations and information during the COVID-19 pandemic is still not sufficiently covered in my view. As such, I find that the authors have not sufficiently addressed my first comment in the original review. Since the authors frame it as a European citizen perspective, they should not limit their literatur to the four countries investigated but take a broader view, at least on the European level. For example, publications that speak to the findings of this study and are able to contrast and refine the interpretation of them might be found in the German COSMO study (https://projekte.uni-erfurt.de/cosmo2020/web/publications/), the UK COSMO study (https://cosmostudy.uk/publications) or the pan-European SolPan project (https://digigov.univie.ac.at/projects/solidarity-in-times-of-a-pandemic-solpan/solpan-publications/). They report on citizen compliance, information behavior, trust in governments, etc. There is much more research published in addition to these three research initiatives from various European contexts and settings.

2. I understand why the authors decided to stratify the focus groups on age, but the emphasis on the age gives the impression that the focuis groups reflect what people that age in that country generally think – which is not possible, given that 3-10 participants are hardly representative of a country’s age group. A more neutral framing of the focus groups (e.g., “Focus group 1, focus gropu 2”) might be helpful to avoid this flawed impression. (I do acknowledge that in some instances, the age matters, e.g. the criticism from young people that students were not sufficiently listened to. But I think it would be enough if the age group was mentioned in these rare occasions only).

3. Relatedly, I am missing some additional information on the study participants to estimate a little bit how well they represent the general citizen. Some demographic characteristics – e.g., how well participants were educated, whether they had children, whether they lived in a rural or urban setting – are highly influential on how they might speak about and value information and whether and how they trust politics.

4. The results present each focus group as a homogeneous entity with similar viewpoints and attitudes. Differences were only reported between age groups or between countries. Weren’t there any relevant discrepancies, conflicts, or differing viewpoints within the focus groups? And if not, why? Too homogeneous viewpoints would, again, raise my concern that the selection of participants is somewhat skewed towards a certain type of citizen. This is hardly avoidable in a qualitative study, but any biases in participant selection should be carefully analysed and reported.

5. The aim of the study is, according to the authors, to “explore the European citizen’s perception of its role in the COVID-19 preparedness, response and recovery” also for future pandemic. However, the authors do not distinguish between preparedness, response and recovery in their results or discussion. This is problematic, since citizen participation might take different forms in times of acute crisis as opposed to more “normal” times.

6. Line 550: “Rather citizens seem to see a more passive role for themselves.” I find this statement problematic because it overly generalises and sees citizens as a homogeneous mass with similar interests and attitudes. It is also not a suitable interpretation of this study’s findings, in my view. If anything, the study shows how citizen perceptions on their own role in policymaking differ, with some wanting more active and others more passive roles. This should be made more nuanced, which is – in my view – the whole point in conducting qualitative studies like this one.

7. The results on trust and the observed country differences should be framed more cautiously, since the study participants might not represent the majority views in each country. The authors should find national polls on citizens’ trust in politics to back up their findings and clarify whether these are representative (e.g. whether they confirm that the Spanish and Slovenian populations are less trusting their governments than Dutch and Finnish populations).

I hope these comments will help the authors to improve their paper further. The methodology seems solid and is well explained and the rationale of the study is clear and relevant. But the relatively few participants from that many different countries poses a challenge to focus the results and discussion to speak meaningfully to the aim of the paper. I also think the authors could leverage more on their qualitative study design to add nuance to citizen perspectives rather than overly (and invalidly) generalise their findings.

7. PLOS authors have the option to publish the peer review history of their article (what does this mean? ). If published, this will include your full peer review and any attached files.

**Do you want your identity to be public for this peer review?** For information about this choice, including consent withdrawal, please see our Privacy Policy .

Reviewer #1: **Yes: ** David Rodríguez-Arias

Reviewer #2: **Yes: ** Bettina M. Zimmermann

---

## [Author Response · Author response to Decision Letter 2]

4 Dec 2024

Dear Prof. Schicktanz,

We would like to thank you and the reviewers once again for providing us with feedback to improve the quality of our manuscript, entitled “The perceptions amongst European citizens of shared responsibility during the COVID-19 management: A focus group study across four European countries”.

The research team has carefully considered and discussed the new comments. Below, you will find our point-by-point responses to all comments, and corresponding changes in the manuscript and where to find them. All significant changes have been marked. All authors have seen and approved the revised manuscript.

We hope this revised manuscript can be accepted for publication in PLOS ONE.

Yours sincerely,

Sandra Kengne Kamga

On behalf of the all the co-authors

Additional Editor Comments:

Dear authors,

we appreciate your revision efforts, but reviewer 1 has some concerns which I also share. Itthat you describe your study design according to main standards of qualitative focus group research, including giving detailed information about socio-demographics (incl. education, gender, profession, perhaps Covid-infection: affected vs. non.affectedness). Also problematic generalizations as outcome of a qualitative study should be avoided. Please replace Euroipean by naming only te countries you have examined (in the title, abstract, introduction, result and discussion parts). Discuss then more closely your findings for the respectives countries in comparison to other existing qualitative and quantitative studies of these countries (Trust-related issues into health sciences, governance, ). You can also discuss your results of your 4 countries with studies that have been undertaken in other countries but please make sure you compare apples with apples...

As qualitative studies help to buil hypothesis but not generalized sztatement, I also recommend that you apply a more reflective discussion style and develop rather an outlook part, how to test your hypothesis in future studies.

We are looking forward to recieve a revised vrsion of your manuscript

Sincerely, Yours Silke Schicktanz

Thank you for your comments. We have replaced “European” with the names of the four countries included throughout the manuscript. We have also referred to previous relevant literature from the four countries when discussing our results more frequently. Moreover, we have changed the wording of some sentences in the discussion to avoid possible overgeneralizations and we have provided suggestions for future research based on our results.

Unfortunately, we do not have more detailed socio-demographics than what has been provided here. We have, however, now reflected on this in the discussion.

Reviewers' comments:

Reviewer's Responses to Questions

Comments to the Author

1. If the authors have adequately addressed your comments raised in a previous round of review and you feel that this manuscript is now acceptable for publication, you may indicate that here to bypass the “Comments to the Author” section, enter your conflict of interest statement in the “Confidential to Editor” section, and submit your "Accept" recommendation.

Reviewer #1: All comments have been addressed

Reviewer #2: (No Response)

2. Is the manuscript technically sound, and do the data support the conclusions?

Reviewer #1: Yes

Reviseer #2: Partly

3. Has the statistical analysis been performed appropriately and rigorously?

Reviewer #1: Yes

Reviewer #2: N/A

4. Have the authors made all data underlying the findings in their manuscript fully available?

Reviewer #1: (No Response)

Reviewer #2: Yes

5. Is the manuscript presented in an intelligible fashion and written in standard English?

Reviewer #1: (No Response)

Reviewer #2: Yes

6. Review Comments to the Author

Reviewer #1: It seems to me that the authors have taken seriously my remarks and, according to their answer, the manuscript has improved sufficiently to be published.

I really appreciate the effort in providing context for each participant country phase of the pandemic. I also appreciate the details given on the methodology and coding process.

In sum, I think this is a great contribution, that has high value in terms of understanding a substantive question: what kind of expertise and legitimacy has the public in engaging with public-health policy decision-making? This paper analyses this problem in depth and is extremely informative.

I think it should be published.

Thank you for your compliments.

Reviewer #2: Thank you for addressing my comments and suggestions. However, I still have some concerns that I would like to share:

1. The vast amoung of literature on citizen perceptions on regulations and information during the COVID-19 pandemic is still not sufficiently covered in my view. As such, I find that the authors have not sufficiently addressed my first comment in the original review. Since the authors frame it as a European citizen perspective, they should not limit their literatur to the four countries investigated but take a broader view, at least on the European level. For example, publications that speak to the findings of this study and are able to contrast and refine the interpretation of them might be found in the German COSMO study (https://projekte.uni-erfurt.de/cosmo2020/web/publications/), the UK COSMO study (https://cosmostudy.uk/publications) or the pan-European SolPan project (https://digigov.univie.ac.at/projects/solidarity-in-times-of-a-pandemic-solpan/solpan-publications/). They report on citizen compliance, information behavior, trust in governments, etc. There is much more research published in addition to these three research initiatives from various European contexts and settings.

Thank you for your comments. We have now included more relevant literature on citizen perception in Finland, the Netherlands, Slovenia, and Spain in the Discussion section. Following the editor’s suggestion, we no longer frame the outcome as a European citizen perspective but focus on the four countries investigated.

2. I understand why the authors decided to stratify the focus groups on age, but the emphasis on the age gives the impression that the focuis groups reflect what people that age in that country generally think – which is not possible, given that 3-10 participants are hardly representative of a country’s age group. A more neutral framing of the focus groups (e.g., “Focus group 1, focus gropu 2”) might be helpful to avoid this flawed impression. (I do acknowledge that in some instances, the age matters, e.g. the criticism from young people that students were not sufficiently listened to. But I think it would be enough if the age group was mentioned in these rare occasions only).

We understand this concern. However, we have stated in the methods, that the aim of the recruitment is to hold an in-depth, explorative discussion with participants across different age categories, rather than to yield a representation sample of the population within the individual countries. We have again emphasized this in the strengths and limitations section.

3. Relatedly, I am missing some additional information on the study participants to estimate a little bit how well they represent the general citizen. Some demographic characteristics – e.g., how well participants were educated, whether they had children, whether they lived in a rural or urban setting – are highly influential on how they might speak about and value information and whether and how they trust politics.

Unfortunately, we do not have more detailed socio-demographics than what has been provided here. We have, however, now extensively reflected on this in the discussion.

4. The results present each focus group as a homogeneous entity with similar viewpoints and attitudes. Differences were only reported between age groups or between countries. Weren’t there any relevant discrepancies, conflicts, or differing viewpoints within the focus groups? And if not, why? Too homogeneous viewpoints would, again, raise my concern that the selection of participants is somewhat skewed towards a certain type of citizen. This is hardly avoidable in a qualitative study, but any biases in participant selection should be carefully analysed and reported.

The aim of this study was to identify key themes, and the differences identified were between the age groups and countries as has been documented. Although not all topics was not evoked or agreed upon by all individuals in the focus groups, we did not omit any explicit no oppositional opinions in our results. We have at times changed the wording of some sentences in the Results and Discussion section, to avoid potential overgeneralization. We have aimed to highlight that certain themes were discussed, rather than suggesting that all participants in particular focus groups had the same opinion. We also have now explicitly acknowledged a possible selection bias of participants in the strengths and limitations section of the Discussion. However, we also feel the probability of this bias has been minimized by the fact that the points of views were not homogenous across age groups or countries.

5. The aim of the study is, according to the authors, to “explore the European citizen’s perception of its role in the COVID-19 preparedness, response and recovery” also for future pandemic. However, the authors do not distinguish between preparedness, response and recovery in their results or discussion. This is problematic, since citizen participation might take different forms in times of acute crisis as opposed to more “normal” times.

Thank you for this remark. Participants were explicitly asked questions related to their roles during preparedness, response, and recovery. Although most of the (sub-) themes identified focused on the response phase, the manuscript also elaborates on some sub-themes specific for the preparedness and recovery phases, under the heading “the citizen’s actions during preparedness and response/recovery.”

6. Line 550: “Rather citizens seem to see a more passive role for themselves.” I find this statement problematic because it overly generalises and sees citizens as a homogeneous mass with similar interests and attitudes. It is also not a suitable interpretation of this study’s findings, in my view. If anything, the study shows how citizen perceptions on their own role in policymaking differ, with some wanting more active and others more passive roles. This should be made more nuanced, which is – in my view – the whole point in conducting qualitative studies like this one.

Thank you for your remark. We acknowledge this statement may have been an overgeneralization. This is the reason why we had already removed it in the previously revised manuscript. This statement is no longer in the most recent manuscript.

7. The results on trust and the observed country differences should be framed more cautiously, since the study participants might not represent the majority views in each country. The authors should find national polls on citizens’ trust in politics to back up their findings and clarify whether these are representative (e.g. whether they confirm that the Spanish and Slovenian populations are less trusting their governments than Dutch and Finnish populations).

We have now provided additional studies and national polls which back up our findings. These can be found in the Discussion.

I hope these comments will help the authors to improve their paper further. The methodology seems solid and is well explained and the rationale of the study is clear and relevant. But the relatively few participants from that many different countries poses a challenge to focus the results and discussion to speak meaningfully to the aim of the paper. I also think the authors could leverage more on their qualitative study design to add nuance to citizen perspectives rather than overly (and invalidly) generalise their findings.

Thank you for your comments. We hope the changes we have made to the manuscript have highlighted both the preliminary and relevant nature of our qualitative results.

7. PLOS authors have the option to publish the peer review history of their article (what does this mean?). If published, this will include your full peer review and any attached files.

Do you want your identity to be public for this peer review? For information about this choice, including consent withdrawal, please see our Privacy Policy.

Reviewer #1: Yes: David Rodríguez-Arias

Reviewer #2: Yes: Bettina M. Zimmermann

While revising your submission, please upload your figure files to the Preflight Analysis and Conversion Engine (PACE) digital diagnostic tool, https://pacev2.apexcovantage.com/. PACE helps ensure that figures meet PLOS requirements. To use PACE, you must first register as a user. Registration is free. Then, login and navigate to the UPLOAD tab, where you will find detailed instructions on how to use the tool. If you encounter any issues or have any questions when using PACE, please email PLOS at figures@plos.org. Please note that Supporting Information files do not need this step

---

## [Decision Letter · Decision Letter 2]

13 Mar 2025

PONE-D-24-07236R2The citizen’s perception of a shared responsibility during the COVID-19 management: Insights from a focus group study across four European countriesPLOS ONE

Dear Dr. Kengne Kamga,

Thank you for submitting your manuscript to PLOS ONE. After careful consideration, we feel that it has merit but does not fully meet PLOS ONE’s publication criteria as it currently stands. Therefore, we invite you to submit a revised version of the manuscript that addresses the points raised during the review process.

**Please double check Figure 1 and correct as necessary. It appears as though Figure 1 is meant to report COVID-19 cases in the four countries, but it currently shows COVID-19 deaths (the same as Figure 2).

We look forward to receiving your revised manuscript.

Kind regards,

Helen Howard

Staff Editor

PLOS ONE

Journal Requirements:

Additional Editor Comments:

Please double check Figure 1 and correct as necessary. It appears as though Figure 1 is meant to report COVID-19 cases in the four countries, but it currently shows COVID-19 deaths (the same as Figure 2).

Reviewers' comments:

Reviewer's Responses to Questions

**Comments to the Author**

1. If the authors have adequately addressed your comments raised in a previous round of review and you feel that this manuscript is now acceptable for publication, you may indicate that here to bypass the “Comments to the Author” section, enter your conflict of interest statement in the “Confidential to Editor” section, and submit your "Accept" recommendation.

Reviewer #1: All comments have been addressed

Reviewer #2: All comments have been addressed

2. Is the manuscript technically sound, and do the data support the conclusions?

Reviewer #1: Yes

Reviewer #2: Yes

3. Has the statistical analysis been performed appropriately and rigorously? 

Reviewer #1: Yes

Reviewer #2: N/A

4. Have the authors made all data underlying the findings in their manuscript fully available?

Reviewer #1: Yes

Reviewer #2: Yes

5. Is the manuscript presented in an intelligible fashion and written in standard English?

Reviewer #1: Yes

Reviewer #2: Yes

6. Review Comments to the Author

Reviewer #1: I appreciate the authors' effort in providing context or each participant country phase of the pandemic. I also appreciate the details given on the methodology and coding process.

In sum, I think this is a great contribution, that has high value in terms of understanding a substantive question: what kind of expertise and legitimacy has the public in engaging with public-health policy decision-making? This paper analyses this problem in depth and is extremely informative.

I think it should be published.

Reviewer #2: (No Response)

7. PLOS authors have the option to publish the peer review history of their article (what does this mean? ). If published, this will include your full peer review and any attached files.

**Do you want your identity to be public for this peer review?** For information about this choice, including consent withdrawal, please see our Privacy Policy .

Reviewer #1: No

Reviewer #2: **Yes: ** Bettina M. Zimmermann

---

## [Author Response · Author response to Decision Letter 3]

13 Mar 2025

Dear Ms. Howard,

We would like to thank you and the reviewers for reviewing our manuscript, entitled “The perceptions amongst European citizens of shared responsibility during the COVID-19 management: A focus group study across four European countries”.

We acknowledge the error that we had made in Figure 1 and have changed the figure accordingly. Furthermore, we have made minor changes in the references list, as we have added several hyperlinks to indicate where some of the references could be found. These changes have been marked in the manuscript.

We hope this revised manuscript can be accepted for publication in PLOS ONE.

Yours sincerely,

Sandra Kengne Kamga

On behalf of the all the co-authors

---

## [Editor Report · Decision Letter 3]

17 Mar 2025

The citizen’s perception of a shared responsibility during the COVID-19 management:

Insights from a focus group study across four European countries

PONE-D-24-07236R3

Dear Dr. Kengne Kamga,

We’re pleased to inform you that your manuscript has been judged scientifically suitable for publication and will be formally accepted for publication once it meets all outstanding technical requirements.

Kind regards,

Helen Howard

Staff Editor

PLOS ONE
---

## [Editor Report · Acceptance letter]

PONE-D-24-07236R3

PLOS ONE

Dear Dr. Kengne Kamga,

I'm pleased to inform you that your manuscript has been deemed suitable for publication in PLOS ONE. Congratulations! Your manuscript is now being handed over to our production team.

Kind regards,

on behalf of

Dr Helen Howard

Staff Editor

PLOS ONE